# Transcriptional pausing at the translation start site operates as a critical checkpoint for riboswitch regulation

Adrien Chauvier[1], Frédéric Picard-Jean[1], Jean-Christophe Berger-Dancause[1], Laurène Bastet[1], Mohammad Reza Naghdi[2], Audrey Dubé[1], Pierre Turcotte[1], Jonathan Perreault[2] & Daniel A. Lafontaine[1]

On the basis of nascent transcript sequencing, it has been postulated but never demonstrated that transcriptional pausing at translation start sites is important for gene regulation. Here we show that the *Escherichia coli* thiamin pyrophosphate (TPP) *thiC* riboswitch contains a regulatory pause site in the translation initiation region that acts as a checkpoint for *thiC* expression. By biochemically probing nascent transcription complexes halted at defined positions, we find a narrow transcriptional window for metabolite binding, in which the downstream boundary is delimited by the checkpoint. We show that transcription complexes at the regulatory pause site favour the formation of a riboswitch intramolecular lock that strongly prevents TPP binding. In contrast, cotranscriptional metabolite binding increases RNA polymerase pausing and induces Rho-dependent transcription termination at the checkpoint. Early transcriptional pausing may provide a general mechanism, whereby transient transcriptional windows directly coordinate the sensing of environmental cues and bacterial mRNA regulation.

[1] Department of Biology, Faculty of Science, RNA Group, Université de Sherbrooke, Sherbrooke, J1K 2R1 Quebec, Canada. [2] INRS-Institut Armand-Frappier, Laval, H7V 1B7 Quebec, Canada. Correspondence and requests for materials should be addressed to D.A.L. (email: Daniel.Lafontaine@USherbrooke.ca).

Gene expression is a highly regulated process that is vital for cellular homoeostasis. Numerous cellular factors such as RNA molecules regulate gene expression at the transcriptional level and contribute to an ever expanding repertoire of mechanisms governing cellular processes[1]. Riboswitches are non-coding RNA elements that directly monitor the level of cellular metabolites and control gene expression by modulating either messenger RNA (mRNA) levels or the efficiency of translation initiation[2]. These RNA switches consist of two modular domains: an aptamer and an expression platform. While the aptamer is implicated in specific metabolite recognition, the expression platform regulates gene expression by controlling the formation of structures that modulate transcription terminators, ribosome-binding sites (RBS), RNase E cleavage sites or mRNA splicing[2]. Recently, identified RNA motifs with unconventional expression platforms[3,4] suggests the existence of novel riboswitch regulation mechanisms not yet discovered.

Although only a few riboswitch regulation mechanisms have been characterized in vivo, growing evidence indicates that multiple biological processes can be controlled by riboswitch conformational changes. For example, it was recently demonstrated that the lysine-dependent lysC riboswitch controls both translation initiation and RNase E cleavage activity on ligand binding[5]. Furthermore, in addition to modulating translation initiation, the ribB and mgtA riboswitches were shown to regulate transcription termination by using the transcription factor Rho[6]. The mgtA riboswitch was also found to contain a critical RNA polymerase pause site important for Rho transcription termination[7]. These results suggest that riboswitch sequences embed all the information required for metabolite sensing and genetic regulation. However, the lack of clear consensus sequences for RNase E or Rho-dependent regulation makes it difficult to predict whether these control mechanisms are widely used by riboswitches[8,9]. Nevertheless, the large substrate recognition spectrum of such RNA binding proteins indicates that they may be involved as key players in the regulation of several riboswitch-controlled genes.

Nascent elongating transcripts sequencing recently revealed a 16-nt consensus pause sequence that is enriched at translation start sites in both E. coli and Bacillus subtilis[10], suggesting that transcriptional pausing may be important for the control of translation initiation. Here we investigate the in vivo regulation mechanism of the thiamin pyrophosphate (TPP)-sensing thiC riboswitch in E. coli. We identify a transcriptional pause site occurring within the vicinity of the translation start codon. Using in vitro assays, we find that pause efficiency increases on TPP binding and is required to trigger Rho-dependent transcription termination. Mutational analysis shows transcriptional pausing to be crucial for mRNA control in vivo. Furthermore, we perform biochemical probing on transcription elongation complexes (ECs) to directly monitor TPP binding on nascent riboswitch transcripts. We find that while TPP binding is efficient for transcription complexes located upstream of the pause site, it is severely reduced for complexes stalled at the pause site. Our results suggest that the nascent RNA forms an intramolecular lock that both prevents TPP binding to the aptamer and exposes the RBS/AUG start codon, promoting translation initiation. The pause site acts as an early transcriptional checkpoint, where a decision is made to either repress or allow transcription elongation and translation initiation of the thiC operon. Here we describe a riboswitch-based molecular mechanism, in which transcriptional pause site is involved in both sensing environmental cues and controlling gene expression.

## Results

**The thiC riboswitch controls transcription and translation.** The E. coli thiC riboswitch is located upstream of the thiCEFSGH operon involved in TPP biosynthesis[11]. Despite the metabolic importance of TPP, no promoter regulation has been reported to modulate thiC expression. However, the thiC riboswitch was shown to negatively regulate gene expression in TPP-rich media[12], ensuring TPP homoeostasis. The riboswitch consists of a conserved aptamer and a large expression platform containing the RBS and AUG start codon (Fig. 1a)[13]. In the TPP-bound conformation (OFF state), the RBS and start codon are sequestered within a stem–loop structure predicted to inhibit translation initiation (Fig. 1a). Currently, no regulation mechanism has been established to explain how the riboswitch regulates gene expression[12], mainly because the complete structure of the ligand-free riboswitch is still unknown. To predict the secondary structure of the thiC riboswitch in the absence of TPP, we retrieved intergenic sequences located upstream of thiC in proteobacteria. By aligning 77 sequences, we identified a highly conserved secondary structure describing the TPP-free ON state (Fig. 1b; Supplementary Fig. 1a,b), in which an alternative pairing (anti-P1 stem) prevents both P1 and RBS-sequestering stems (Fig. 1a), suggesting efficient translation initiation. In this model, the ON state structure of the thiC riboswitch is mutually exclusive to the TPP-bound conformer, thereby providing a molecular mechanism explaining how the thiC riboswitch modulates translation initiation on TPP binding.

We investigated the thiC riboswitch regulatory mechanism using in vivo reporter gene assays. However, since the transcription start site (TSS) was unknown, we first determined the transcriptional $+1$ residue by performing primer extension analysis on thiC mRNA (Supplementary Fig. 2a). The deduced mRNA sequence differed from previously used transcript versions[12] and the newly determined TSS allows formation of an elongated P1 stem, including residues 1–6 and 91–95 (Fig. 1a). We engineered transcriptional and translational lacZ chromosomal E. coli constructs comprised of the thiC riboswitch sequence and the first 10 codons of thiC open reading frame (ORF). While translational fusions report on protein and mRNA levels, transcriptional thiC–lacZ fusions have two independent RBS (thiC and lacZ) allowing us to monitor only thiC mRNA levels (Supplementary Fig. 2b). The β-galactosidase activity was repressed by 65% in the presence of TPP in the context of the translational fusion (Fig. 1c, WT, trL construct). TPP-induced gene repression was also observed for the transcriptional fusion (Fig. 1c, WT, trX construct), in agreement with the riboswitch modulating thiC mRNA levels[12]. To confirm that the genetic repression is controlled through riboswitch conformational changes, we tested constructs in which the riboswitch P1 stem was stabilized in the OFF conformation to negatively regulate thiC expression (Supplementary Fig. 2c). As expected, β-galactosidase activities were significantly reduced and were not affected by the presence of TPP (Fig. 1c). The introduction of a single-point mutation (G31C) preventing ligand binding[14] resulted in the loss of TPP-dependent gene repression (Fig. 1c, G31C construct), indicating that riboswitch metabolite binding is required for genetic regulation.

**The thiC riboswitch modulates Rho-dependent termination.** As previously observed for E. coli riboswitches, we speculated that TPP-dependent modulation of thiC mRNA levels could result from either RNase E-dependent mRNA decay[5] or Rho-dependent transcription termination[6]. To establish whether RNase E regulates thiC mRNA levels, we used the bacterial strain rne-131 that prevents RNA degradosome assembly[15]. An efficient TPP-mediated regulation was maintained using thiC–lacZ

transcriptional fusions in the *rne-131* strain (Fig. 1d), suggesting that RNase E is not involved in the control of *thiC* mRNA levels. However, we observed a complete loss of TPP-mediated gene regulation in the presence of the Rho inhibitor bicyclomycin[16] (Fig. 1e), suggesting that Rho regulates *thiC* mRNA levels. We then determined the position of Rho-dependent transcription termination, using single-round *in vitro* transcription assays[6]. Control transcription reactions performed in the absence of Rho did not produce any prematurely terminated products, consistent with the absence of intrinsic transcription termination (Fig. 1f, see Ctrl lanes). However, TPP-induced transcription termination (~16%) was detected in presence of Rho (Fig. 1f, see Rho lanes). Transcriptional sequencing revealed that the termination position corresponds to C187 (Supplementary Fig. 3a). Further addition of the Rho-interacting factor NusG[17,18] yielded significantly stronger TPP-dependent termination (~64%) at both positions U184 and C187 (Fig. 1f). Together, these results indicate that TPP binding to the *thiC* riboswitch results in Rho-dependent transcription termination at positions U184 and C187.

We next sought to identify the Rho-binding site in the riboswitch expression platform. Rho-binding sequences (*rut* sites) are usually rich in cytosine but poor in guanine, and are relatively unstructured[19]. Close inspection of *thiC* nucleotide composition indicated that the 100–140 nt region is a good match for a typical *rut* site (Supplementary Fig. 3b). Partial S1 nuclease digestion on nascent *thiC* RNA showed that positions 102–104 and 137–138 are more accessible in the presence of TPP (Supplementary Fig. 3c), suggesting that they are targeted by Rho in the ligand-bound state. Furthermore, we observed an increase in nuclease digestion of these regions in presence of Rho (Supplementary Fig. 3d), indicating that Rho binding reorganizes the riboswitch structure. Our data are consistent with Rho crystal structures, in which RNA-binding sites are proposed to allow the RNA to wrap around the Rho hexamer[20,21]. Interestingly, Rho binding to the *mgtCBR* leader also increases nuclease digestion of the *rut* site[22], suggesting that both *thiC* and *mgtCBR* sequences share similar Rho-binding mechanisms. Thus, these results indicate that TPP binding modulates access to a *rut* site located within the *thiC* riboswitch expression platform.

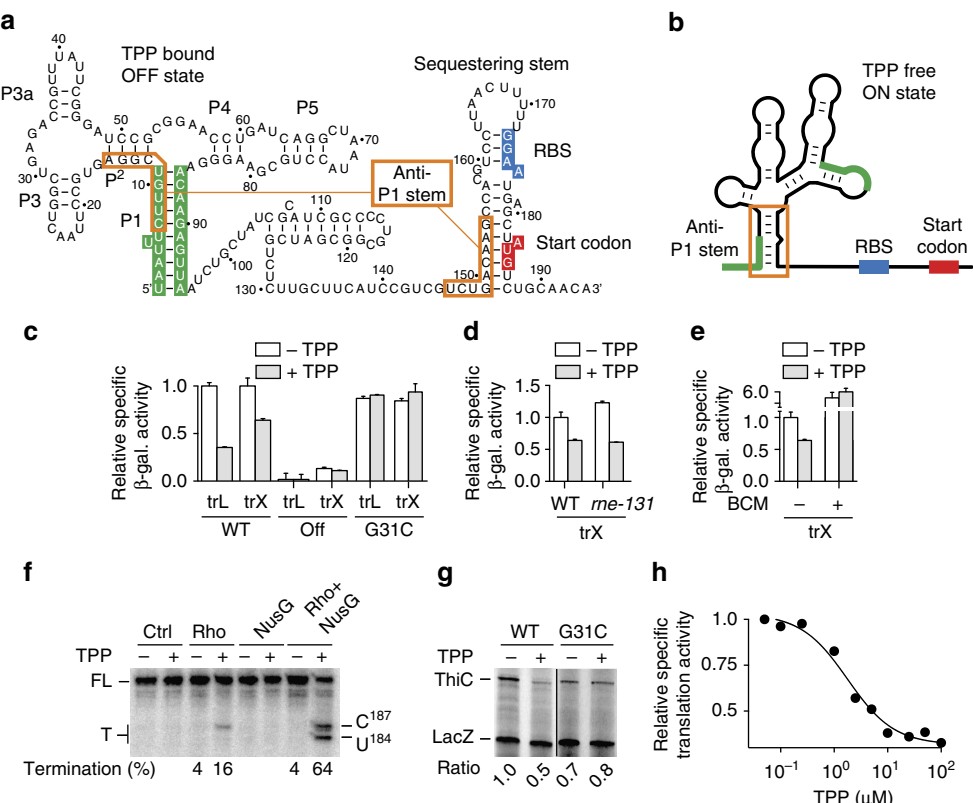

**Figure 1 | The *thiC* riboswitch controls gene expression at the levels of transcription and translation.** (**a**) Secondary structure of the *thiC* riboswitch representing the TPP-bound state. The P1 stem, the RBS and the start codon are represented in green, blue and red, respectively. The anti-P1 stem is shown in orange rectangles. The nomenclature of the helical domains is based on a previous study[12]. (**b**) Schematic representing the TPP-free state determined in this study. The colour scheme is the same as in **a**. (**c**) β-Galactosidase assays of translational ThiC–LacZ (trL) and transcriptional *thiC–lacZ* (trX) fusions for the wild type, ON, OFF and G31C mutants. Values were normalized to the activity obtained for the wild type in the absence of TPP. The average values of three independent experiments with s.d.'s are shown. (**d**) β-Galactosidase assays of transcriptional *thiC-lacZ* (trX) performed in the context of the wild-type and *rne-131* strains. Values were normalized to the activity obtained for the wild type in absence of TPP. The average values of three independent experiments with s.d.'s are shown. (**e**) β-Galactosidase assays of transcriptional *thiC–lacZ* (trX) performed in the wild type in absence or presence of TPP or bicyclomycin (BCM). Values were normalized to the activity obtained for the wild type in absence of TPP. The average values of three independent experiments with s.d.'s are shown. (**f**) In vitro Rho-dependent transcription performed using the *thiC* riboswitch. Full-length (FL) and termination (T) products are indicated on the left. Termination efficiencies are indicated below for reactions done with Rho. Termination products are indicated on the right. (**g**) Transcription–translation-coupled *in vitro* assays using the wild type and G31C mutant. Reactions were performed in absence ( − ) or presence ( + ) of TPP. Translation products are shown on the left and ratios of ThiC expression are normalized to that of LacZ. (**h**) Transcription–translation-coupled *in vitro* assays using the wild-type riboswitch. The half transition of translation repression corresponds to a value of $1.7 \pm 0.3 \mu M$.

**The *thiC* riboswitch directly controls translation initiation.** On the basis of the secondary structure model, it is predicted that the *thiC* riboswitch modulates the initiation of translation on TPP binding (Fig. 1a). To directly monitor TPP-dependent control of *thiC* translation initiation, we carried out transcription–translation-coupled *in vitro* assays using the cell-free PURExpress system[23]. As an internal control, we included a DNA template corresponding to a *lacZ* fragment in the same reaction. In the absence of TPP, ThiC and LacZ translation products were detected (Fig. 1g, WT, − lane). However, we observed a twofold reduction of ThiC products in the presence of TPP, consistent with the riboswitch-repressing translation initiation on ligand binding (Fig. 1g, WT, + lane). No such TPP-dependent repression was observed when using a G31C mutant riboswitch (Fig. 1g, G31C), in agreement with *in vivo* data (Fig. 1c). Using the wild-type construct, we next performed *in vitro* translation assays while varying the TPP concentration. Fitting analysis revealed a half transition value of $1.7\,\mu M \pm 0.3\,\mu M$ for translation repression (Fig. 1h). Given that the *in vivo* TPP concentration is in the low micromolar range[24], it suggests that the *thiC* riboswitch is well tuned to detect relevant cellular TPP variations in *E. coli*.

**TPP binding induces riboswitch conformational rearrangements.** To investigate how TPP modulates the structure of the *thiC* riboswitch, we used the well-established RNase H cleavage assay that was previously used to investigate the conformation of nascent *E. coli btuB* riboswitch mRNAs transcribed by *E. coli* RNA polymerase[25,26]. We used a similar approach in which we designed 10-mer DNA oligonucleotide probes to target the P1 stem (positions 10–19) and the RBS region (positions 172–181) (Fig. 2a). Nascent RNAs were first studied using the P1 probe in the absence or presence of TPP. While a strong cleavage product was observed in the absence of TPP (Fig. 2a), a complete loss of RNase H cleavage was obtained in presence of the ligand (Fig. 2a), consistent with TPP stabilizing the formation of the P1 stem (Fig. 1a). We obtained similar results using a DNA oligonucleotide targeting the RBS domain (Fig. 2a), indicating that the RBS-sequestering stem is also stabilized on TPP binding.

Next, we performed RNase H assays to study the TPP-induced *thiC* riboswitch conformational change as a function of TPP concentration. The cleavage efficiency of RNase H targeting the P1 stem progressively decreased as the TPP concentration increased (Fig. 2b; Supplementary Fig. 4a). Fitting analysis yielded a half TPP concentration ($K_{switch}$) value of $422\,nM \pm 27\,nM$ for the structural modulation. A similar $K_{switch}$ value of $453\,nM \pm 13\,nM$ was obtained using the RBS probe (Fig. 2b; Supplementary Fig. 4b). Given that both $K_{switch}$ values obtained using the P1 and RBS probes are approximately fourfold higher than the 100 nM dissociation constant of the TPP-*thiC* aptamer complex[12], it suggests that the *thiC* riboswitch operates under a kinetic regime where the transcription rate is important for ligand binding, as reported for *B. subtilis* flavin mononucleotide[27] and adenine[28] riboswitches. As expected for a kinetic regime, we found that $K_{switch}$ values varied with decreasing nucleotide triphosphate (NTP) concentrations (Supplementary Table 1), consistent with slower transcription rates allowing riboswitch regulation at lower ligand concentrations[27,28].

**The *thiC* riboswitch performs TPP sensing cotranscriptionally.** To investigate the kinetics of TPP binding, we assessed the efficiency of metabolite sensing in a cotranscriptional or post-transcriptional context using RNase H assays. In these experiments (Fig. 2c), TPP was added either cotranscriptionally or post transcriptionally (after the addition of heparin). As expected, control experiments showed that the RBS region is accessible to RNase H cleavage in the absence of the ligand (Fig. 2d, Ctrl, + lane). Cotranscriptional binding of TPP strongly protected the RBS, even at the first time point (Fig. 2d, Co-trX, 15 s). However, post-transcriptional TPP addition yielded weaker RNase H protection for early time points (15, 45 and 90 s; Fig. 2d, Post-trX), suggesting a slower rate of TPP binding. Fitting analysis yielded a fast-binding rate having a lower bound of $0.20 \pm 0.04\,s^{-1}$ for cotranscriptional TPP binding and an apparent binding rate of $0.05 \pm 0.01\,s^{-1}$ for post-transcriptional binding (Supplementary Fig. 5a), consistent with cotranscriptional

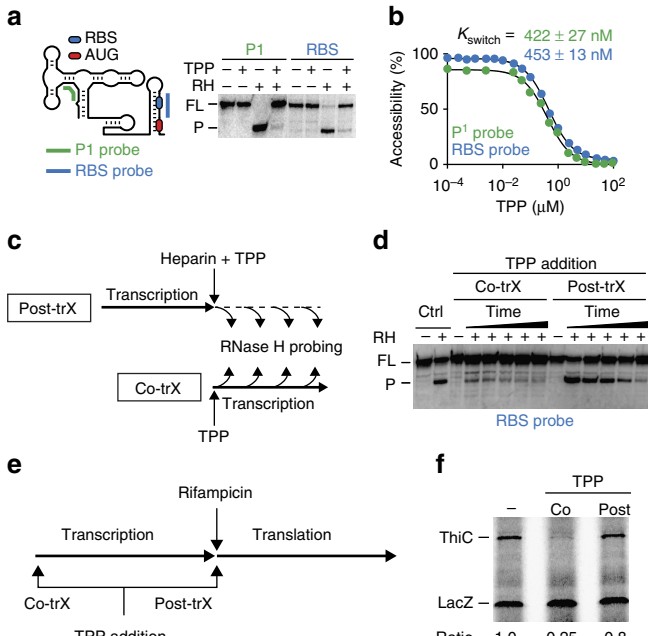

**Figure 2 | The *thiC* riboswitch performs TPP sensing cotranscriptionally.** (**a**) Schematic of the construct used for RNase H probing assays and the locations of P1 (green) and RBS (blue) probes. The right panel shows RNase H assays performed in the absence (−) or presence (+) of TPP or RNase H (RH), using the P1 or RBS probe. The full-length (FL) and cleaved product (P) are shown on the left. (**b**) $K_{switch}$ determination for the wild-type riboswitch. *In vitro* transcription reactions were done in presence of increasing TPP concentrations ranging from 100 pM to 100 μM. Experiments were done in triplicate and the data shown are a representative result. (**c**) Experimental setup monitoring TPP binding to the *thiC* riboswitch. Post-transcriptional TPP binding (Post-trX) was assessed by performing transcription in absence of TPP. After stopping transcription, TPP is added and RNase H assays are performed at several time points. Cotranscriptional TPP binding (Co-trX) was assessed by adding TPP before transcription initiation and by performing RNase H assays during transcription. (**d**) RNase H probing assays monitoring TPP binding to the *thiC* riboswitch. Control reactions (Ctrl) done in presence of RNase H (RH) show a cleaved product (P) without TPP. Post-transcriptional and cotranscriptional reactions were performed as indicated in **c**. The FL and P are indicated on the left. (**e**) Experimental setup monitoring TPP binding using transcription–translation *in vitro* assays. The effect of post-transcriptional (Post-trX) TPP binding was assessed by initiating the transcription reaction in absence of TPP. After the addition of rifampicin, amino acids and TPP were added to allow TPP binding to the riboswitch during translation. The cotranscriptional (Co-trX) TPP binding was assessed by adding TPP at the beginning of transcription.
(**f**) Transcription–translation *in vitro* assays assessing TPP binding to the riboswitch. Reactions were performed in absence (−) or presence of TPP added either cotranscriptionally (Co) or post transcriptionally (Post) as indicated in **e**. Ratios of ThiC expression normalized to that of LacZ.

ligand binding being approximately fourfold more efficient. Similar results were obtained using a P1 probe (Supplementary Fig. 5b,c). Given that rifampicin treatments showed that *thiC* mRNA half-life is extremely short *in vivo* (Supplementary Fig. 5d, <60 s), our results support the idea that the *thiC* riboswitch regulates gene expression in bacteria by binding TPP cotranscriptionally.

We next tested cotranscriptional TPP binding using transcription–translation *in vitro* assays. Here, transcription was uncoupled from translation by omitting the addition of amino acids during the transcription step (Fig. 2e). In the second step, rifampicin and amino acids were added to stop transcription and initiate translation, respectively (Fig. 2e). As shown in Fig. 2f, while the cotranscriptional binding of TPP reduced ThiC expression by approximately fourfold, no effect was observed when TPP was added at the beginning of the translation step (post transcriptionally). These results indicate that the transcription process is a major determinant for both TPP sensing and regulation of translation initiation.

**Pausing in the start codon vicinity is important for control**. The high efficiency of TPP cotranscriptional recognition by the *thiC* riboswitch suggests that metabolite sensing is performed within a defined transcriptional window. Of the key factors involved in the transcription process, RNA polymerase (RNAP) transcriptional pausing has been shown to influence RNA folding in numerous systems[7,25,27–29] and to be important for Rho-dependent transcription termination[7,30]. To determine RNAP pausing within the *thiC* riboswitch, we performed single-round *in vitro* transcription assays and analysed transcripts at various time points (Fig. 3a). Three pause regions were observed in the riboswitch expression platform that encompassed positions A138, C158 and C187 (Supplementary Fig. 6a). Interestingly, not only is RNAP pausing at C187 remarkably efficient compared with both A138 and C158, but also the half-life of the pause is increased by ~2.8-fold in presence of TPP (Fig. 3b; Supplementary Fig. 6b). That the C187 pause half-life is TPP-dependent suggests that the *thiC* riboswitch structure is involved in RNAP pausing. Accordingly, we found that destabilizing the U160-A175 hairpin by introducing a U160A mutation abolished the effect of TPP on the C187 pause site (Fig. 3b; Supplementary Fig. 6c).

Given that Rho-dependent transcription termination occurs at C187 (Fig. 1f), we speculated that RNAP pausing at C187 could be important for transcription termination. To verify this hypothesis, we identified a mutation (U186A) in the *thiC* riboswitch that strongly inhibits TPP-dependent pausing at C187 (Fig. 3b; Supplementary Fig. 7a). We found that the U186A mutation severely impaired Rho transcription termination *in vitro* (Supplementary Fig. 7b), suggesting a critical role for the C187 pause site in Rho termination activity. The U186A mutation did not alter TPP-mediated riboswitch conformational changes when using RNase H (Fig. 3c; Supplementary Fig. 7c,d) or in transcription–translation assays (Fig. 3d). Accordingly, *in vivo* assays revealed that while a U186A translational *lacZ* fusion exhibits regulation similar to the wild-type construct (Fig. 3e, U186A, trL), the efficiency of TPP-dependent mRNA regulation is reduced when using transcriptional lacZ fusions (Fig. 3e, U186A, trX). These results suggest that the *thiC* riboswitch does not rely on Rho transcription termination to regulate at the translational level. Together, *in vitro* and *in vivo* results clearly indicate the critical role of the C187 pause site in controlling Rho transcription termination on TPP binding.

**Monitoring TPP binding in transcription ECs**. To determine how cotranscriptional TPP binding modulates the structure of the *thiC* riboswitch, we developed a structural probing approach using RNAP transcription ECs halted at specific positions. In these experiments, we halted various EC using a biotin–streptavidin roadblock located downstream of the stalled position[31]. For example, RNAP stalling at the first pause site (EC-138) was obtained by introducing a biotin–streptavidin complex at position C147 (Fig. 4a). As expected, the presence of the 138-mer transcript was found to be biotin and streptavidin-dependent (Supplementary Fig. 8a). Furthermore, the 138-mer nascent transcript was still detected after a wash step, in which transcription complexes were attached to cobalt beads via the RNAP His tag (Supplementary Fig. 8b), consistent with the presence of intact transcription ECs.

We first monitored the binding of TPP to EC-138 using RNase H assays. In the absence of TPP, we obtained 75% cleavage efficiency using the P1 probe (Fig. 4b, EC-138), indicating that the P1 stem is mostly accessible under this conditions. However, adding TPP significantly decreased cleavage efficiency (down to 8%), giving a cleavage ratio of 9.4 (Fig. 4b, EC-138). These results suggest that TPP sensing is efficiently performed in the context of EC-138. The efficiency of TPP binding was reduced when using upstream complexes EC-100 and EC-108, giving cleavage ratios of 1.3 and 6.7, respectively (Fig. 4b, EC-100 and EC-108). The strongly impaired TPP sensing ability of the EC-100 complex could be due to RNAP hindering P1 stem formation. However, we observed efficient TPP sensing when RNAP was stalled at the second pause site (Fig. 4b, EC-158). Strikingly, we obtained a markedly reduced cleavage ratio of 2.8 at the third pause site (EC-187), suggesting that TPP binding is not efficient for this transcription complex (Fig. 4b, EC-187). Nevertheless, TPP sensing was partially recovered in complexes located at down-stream ORF positions (EC-197 and EC-207) (Fig. 4b). These results indicate that although metabolite binding is efficient when RNAP is located at the first or second pause site (EC-138 and EC-158), TPP modulation decreases significantly at the third pause site (EC-187).

We measured the TPP-binding affinity of transcription complexes by performing RNase H assays using a range of TPP concentrations. We found that TPP sensing is more potent when RNAP is stalled at pause sites 138 ($28 \pm 1$ nM) or 158 ($27$ nM $\pm 2$ nM) (Fig. 4c; Supplementary Fig. 9a,b) compared with the full-length transcript (Fig. 2b). However, we obtained a markedly increased $K_{switch}$ value of $2.1 \pm 0.3 \, \mu$M for EC-187 (~100-fold higher than EC-138 or EC-158; Supplementary Fig. 9c), indicating that TPP binding is severely impaired at this pause site. The large difference in TPP affinity suggests that the *thiC* nascent transcript in the context of EC-187 folds into a structure different from those adopted in EC-138 and EC-158. Indeed, given that the last 14 transcribed residues (positions 176–187) are protected within RNAP[27,31,32], we expect that the RNAP stalled at position 187 prevents the formation of the RBS-sequestering stem. As a result, the folding of the long-range anti-P1 stem is favoured, locking the aptamer domain into a non-competent TPP-binding structure. This structure cannot be formed in EC-138 or EC-158 since the 3′-moiety of the anti-P1 stem is either not yet transcribed (EC-138) or contained in the RNAP exit channel (EC-158; Fig. 4d). We confirmed the negative influence of the anti-P1 stem on TPP binding by introducing destabilizing anti-P1 stem mutations at positions 147–149 in EC-187 (Supplementary Fig. 9d), where increased TPP-sensing affinity was measured for the mutant ($410$ nM $\pm 53$ nM) in comparison with the wild-type EC-187 complex ($2.1 \pm 0.3 \, \mu$M).

In contrast to EC-187, we speculated that the presence of RNAP in EC-158 would prevent the formation of the anti-P1 stem, thereby positively influencing TPP binding (Fig. 4d). As expected, RNase H probing of an identical 158 nt transcript in the

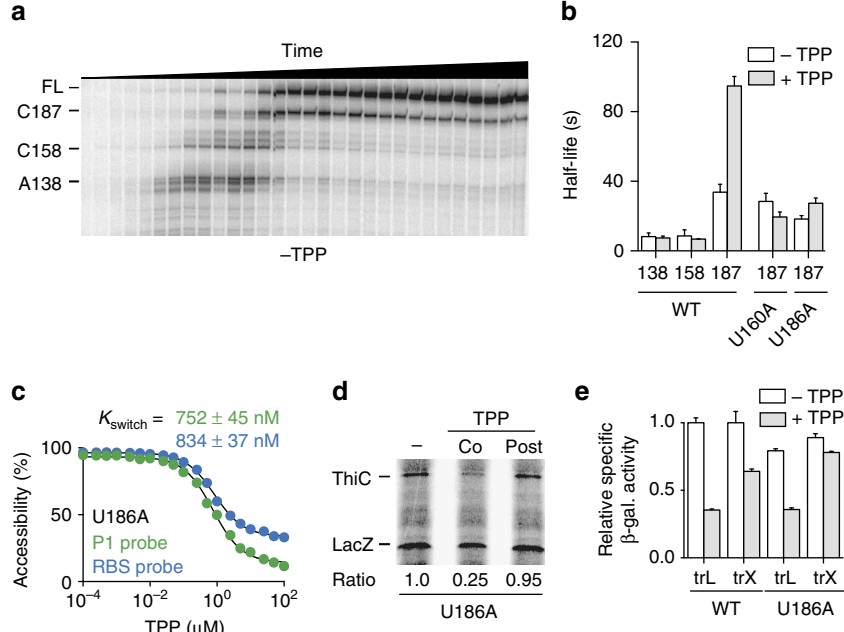

**Figure 3 | The *thiC* riboswitch exhibits transcriptional pausing at the translation start site important for transcriptional control.** (**a**) Transcriptional pausing of the *thiC* riboswitch in the absence of TPP using 25 μM NTP. This nucleotide concentration is used to allow easier detection of transcriptional pausing. Pause sites A138, C158 and C187 are indicated on the left. The mapping of the pauses sites is shown in Supplementary Fig. 6a. (**b**) Quantification of the half-life of transcriptional pause sites in the wild type, and the U160A and U186A mutants. Experiments were performed using 100 μM NTP in absence and presence of 10 μM TPP. Quantification data for pauses A138 and C158 are found in the Supplementary Table 2. The average values of three independent experiments with s.d.'s are shown. (**c**) $K_{switch}$ determination using the P1 or RBS probe for the U186A riboswitch mutant. *In vitro* transcription reactions were done with increasing TPP concentrations ranging from 100 pM to 100 μM. $K_{switch}$ values of 752 ± 45 nM and 834 ± 37 nM were obtained using the P1 and RBS probes, respectively. Experiments were done in triplicate and the data shown are a representative result. (**d**) Transcription–translation *in vitro* assays assessing TPP binding to the U186A riboswitch mutant. Reactions were performed in absence ( − ) or presence of 25 μM TPP added either cotranscriptionally (Co) or post transcriptionally (Post) as indicated in Fig. 2e. Ratios of ThiC expression are indicated below. (**e**) β-Galactosidase assays of translational ThiC–lacZ (trL) and transcriptional *thiC–lacZ* (trX) fusions for the wild type and U186A mutant. Enzymatic activities were determined in absence and presence of 0.5 mg ml$^{-1}$ TPP. Values were normalized to the activity obtained for the wild type in the absence of TPP. The average values of three independent experiments with s.d.'s are shown.

absence of RNAP resulted in less efficient TPP-mediated structural change compared with EC-158 (Fig. 4e, 158 nt, ratio = 3.4), consistent with the anti-P1 stem perturbing TPP sensing. In agreement with this, using a shorter 143 nt transcript that did not allow the anti-P1 stem to form showed an increased cleavage ratio (Fig. 4e, 143 nt). Together, our results support a model in which selective exposure of the 146–154 nt region dictates anti-P1 stem formation, which acts as an intramolecular lock preventing TPP sensing.

**SHAPE analysis of transcription ECs**. RNA structure can be studied at the nucleotide level using selective 2′-hydroxyl acylation analysed by primer extension (SHAPE)[28,33–40]. We adapted this highly informative technique to interrogate the RNA structure in the context of EC-187. We produced nascent *thiC* transcripts using *E. coli* RNAP and either halted them at position 187 (EC-187) or allowed RNAP to transcribe the full riboswitch sequence. Immediately after transcription, we probed the RNA structure using 2-methylnicotinic acid imidazolide (NAI). Interestingly, SHAPE reactivity was similar for the EC-187 nascent RNA and full-length transcript, with the noteworthy exception of the C7-G16 region (Fig. 4f; Supplementary Fig. 10a). The decrease in reactivity for residues C7-G16 in EC-187 suggests that this region is involved in a stable structure, such as the long-range anti-P1 stem interaction (Fig. 1a). However, in absence of TPP, the higher SHAPE reactivity of C7-G16 in the full-length transcript suggests that the anti-P1 stem is not the predominant

form. This is presumably due to an equilibrium shift toward the formation of the RBS-sequestering stem that can form in the context of the full-length riboswitch, but not in the context of EC-187.

We next repeated similar experiments but added TPP cotranscriptionally. For both the EC-187 complex and the full-length transcript, adding NAI resulted in reactivity changes in both the aptamer and expression platform (Supplementary Fig. 10b–d). For each species, most TPP-induced changes observed in the aptamer domain were similar to those reported for the *thiM* riboswitch[39,41]. For example, NAI reactivity increased for positions C53, G54, A56, U63 and A80. The expression platform produced a TPP-dependent reactivity profile consistent with the predicted structure, in which residues U131, U132, C141, G142 and U146 were more reactive in the presence of TPP (Supplementary Fig. 10b–d). Taken together, SHAPE analyses show that, in the context of EC-187, the *thiC* riboswitch adopts a TPP-free structure in which the anti-P1 stem is stabilized.

**Riboswitches exhibit pausing in start codon vicinity**. To determine if transcriptional pausing in the translation start region is widespread among *E. coli* riboswitches, we performed transcription kinetics for the *thiB*, *thiM*, *ribB*, *btuB* and *lysC* riboswitches (Fig. 5; Supplementary Figs 11–15). In all cases, we found a clear pausing region in the vicinity of the RBS and start codon (Fig. 5a). We did not detect a significant

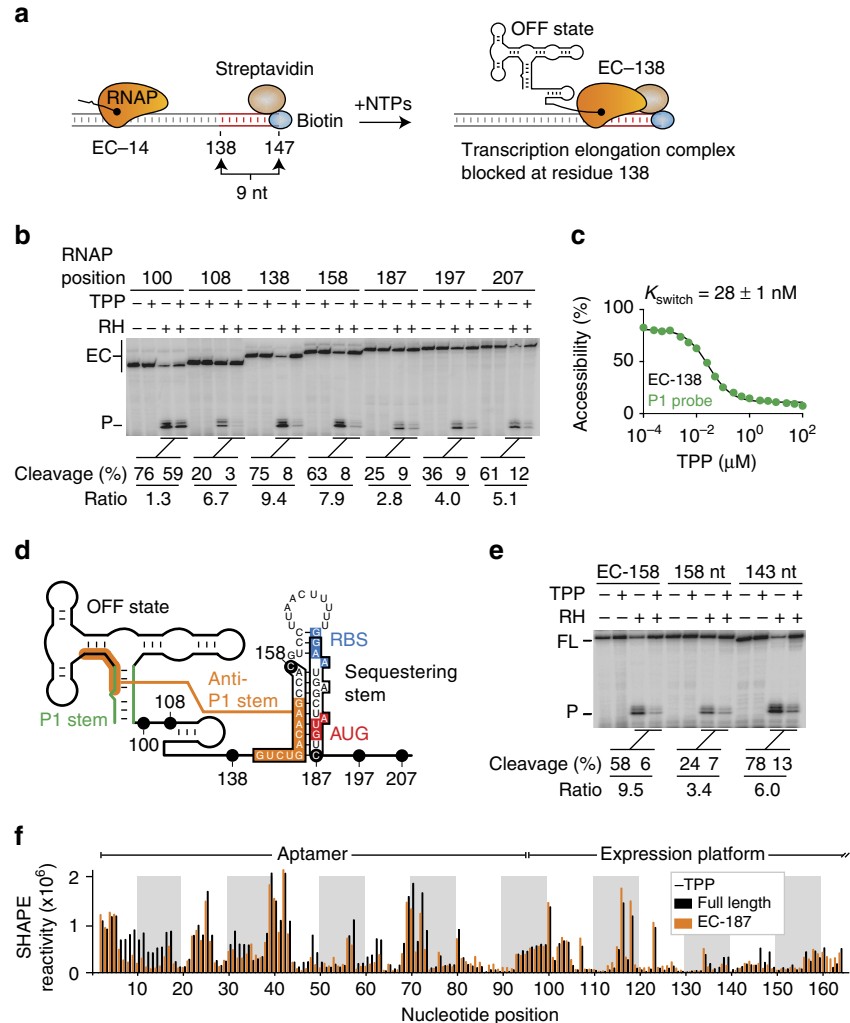

**Figure 4 | The TPP-binding efficiency is modulated along the *thiC* riboswitch transcriptional pathway. (a)** Schematic of the experimental system for the formation of elongation complexes (ECs) at position 138 (EC-138). Transcription initiation performed using a subset of NTP leads RNAP to stall at position 14 (EC-14). Using a DNA template containing a biotin–streptavidin complex, the addition of all four NTP results in RNAP to block at nine residues upstream from the end of the template, thus producing stable EC-138 complexes. **(b)** RNase H probing assays performed on *thiC* riboswitch ECs. Reactions were done in absence ( − ) or presence ( + ) of 10 μM TPP and in the absence ( − ) or presence ( + ) of RNase H (RH). Uncleaved ECs and cleaved products (P) are indicated on the left. Cleavage efficiencies and ratios are indicated below. **(c)** $K_{switch}$ determination using the P1 probe for the EC-138 complex. Cleavage reactions were done with increasing TPP concentrations ranging from 100 pM to 100 μM. A $K_{switch}$ value of 28 nM ± 1 nM was obtained using the P1 probe. The corresponding gel is shown in Supplementary Fig. 9a. Experiments were done in triplicate and the data shown are a representative result. **(d)** Schematic of the *thiC* riboswitch showing the different stalled RNAP positions. For pause sites C158 and C187, the predicted sequence contained in the RNAP exit channel is boxed. The localization of C158 and C187 pause sites allows RNAP to either inhibit or allow the formation of the anti-P1 stem, respectively. **(e)** RNase H probing assays done on the EC-158 EC and on 158 nt and 143 nt *thiC* transcripts. Reactions were done in absence ( − ) or presence ( + ) of 10 μM TPP and in absence ( − ) or presence ( + ) of RNase H (RH). Uncleaved (FL) and cleaved products (P) are indicated on the left. Cleavage efficiencies and ratios are indicated below. **(f)** SHAPE modification of the *thiC* riboswitch performed on the full-length transcript and EC-187 EC in absence of TPP. The histogram represents the relative SHAPE reactivity for positions 1–164.

metabolite-dependent effect on half-life pausing for *btuB, thiM* and *lysC* riboswitches (Fig. 5b–d). However, while adding TPP resulted in a marked reduction in half-life pausing in the region of position 117 of the *thiB* riboswitch (Fig. 5e), the *ribB* riboswitch showed an increased pause half-life in the start region with flavin mononucleotide (Fig. 5f), in agreement with Rho transcription termination[6]. In all cases, the localization of the pause site in the vicinity of the RBS/start codon suggests the formation of an anti-P1 stem by allowing RNAP either to sequester a competing sequence (*thiB, thiM, btuB* and *lysC*) or by delaying the transcription of the anti-P1 competing sequence (*ribB*) (Supplementary Figs 11–15). The predicted anti-P1 stem of the *lysC* riboswitch is particularly large (Supplementary Fig. 13),

suggesting very efficient translation initiation. This is in agreement with *lysC* exhibiting the highest protein synthesis rate in minimal media among *E. coli* riboswitches[42]. Detailed biochemical characterization will be required to elucidate the role of riboswitch pause sites. The presence of multiple pause sites within riboswitch expression platforms is intriguing and strongly suggests that the transcription process is a key element in riboswitch ligand sensing and gene regulation.

## Discussion

Since their discovery, riboswitches have been extensively studied for their importance in bacterial gene regulation and more

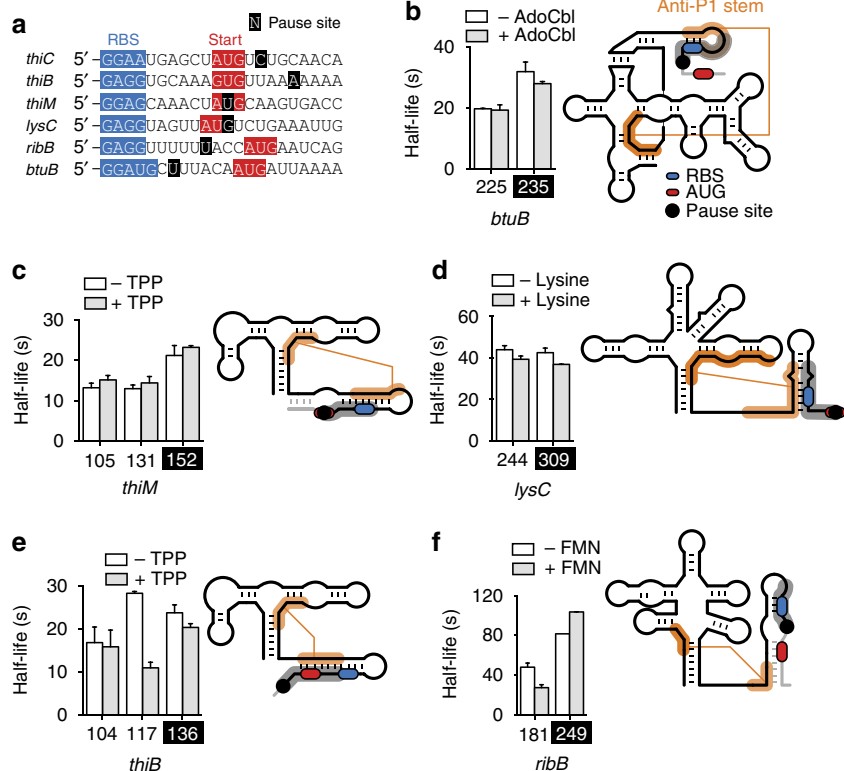

**Figure 5 | Transcriptional pause regions in *E. coli* riboswitches.** (**a**) Sequences of *E. coli* riboswitch start codons. A single position designates each pausing region, centred around positions 187 (*thiC*), 136 (*thiB*), 152 (*thiM*), 309 (*lysC*), 249 (*ribB*) and 235 (*btuB*). Pause sites, RBS and AUG start codon are indicated in black, blue and red, respectively. (**b**) Transcriptional pausing in the *btuB* riboswitch. A schematic representation of the *btuB* riboswitch is shown on the right and the 235 pause site is indicated. The grey region represents the section of the riboswitch transcript sequestered by the RNA polymerase when stalled at the pause site, thus allowing anti-P1 formation (shown in orange lines). The RBS and AUG start codon are represented in blue and red, respectively. The pause regions and riboswitch sequence are shown in Supplementary Fig. 11. (**c**) Transcriptional pausing in the *thiM* riboswitch. A schematic representation of the *thiM* riboswitch is shown on the right and the pause site at residue 152 is indicated. The colour scheme is identical to the one used in **b**. The pause regions and riboswitch sequence are shown in Supplementary Fig. 12. (**d**) Transcriptional pausing in the *lysC* riboswitch. A schematic representation of the *lysC* riboswitch is shown on the right and the pause site at residue 309 is indicated. The colour scheme is identical to the one used in **b**. The pause regions and riboswitch sequence are shown in Supplementary Fig. 13. (**e**) Transcriptional pausing in the *thiB* riboswitch. A schematic representation of the *thiB* riboswitch is shown on the right and the pause site at residue 136 is indicated. The colour scheme is identical to the one used in **b**. The pause regions and riboswitch sequence are shown in Supplementary Fig. 14. (**f**) Transcriptional pausing in the *ribB* riboswitch. A schematic representation of the *ribB* riboswitch is shown on the right and the pause site at residue 249 is indicated. The colour scheme is identical to the one used in **b**. The pause regions and riboswitch sequence are shown in Supplementary Fig. 15. The average values of three independent experiments with s.d. s are shown.

recently as potential targets for novel antimicrobial agents[43,44]. Despite the large amount of structural information obtained for aptamer structures[2], only a few studies have addressed riboswitch regulation mechanisms and how metabolite binding coordinates gene expression. Here we elucidate the regulation mechanism of the *E. coli thiC* riboswitch and show that TPP binding mediates both Rho-dependent transcription termination and repression of translation initiation. We developed an approach in which cotranscriptional TPP-mediated conformational changes were monitored in the context of transcription ECs stalled at specific positions. Our results demonstrate that the *thiC* riboswitch regulates in a narrow transcriptional window in which the downstream boundary is delimited by a critical RNAP pausing site found in the vicinity of the translation initiation codon. Strikingly, while TPP-binding affinity is high for transcription complexes located upstream of the pause site, ligand binding is impaired by ∼100-fold when transcription complexes are stalled at this regulatory checkpoint. In contrast, TPP-bound transcription complexes reaching the checkpoint undergo transcription termination or translation repression. Such a regulatory checkpoint provides a strategic gate where the outcome of the genetic decision is sealed. Given that

transcriptional pausing occurs widely in *E. coli* and *B. subtilis*[10], our study provides a working model describing how riboswitches—or any structured RNA element—may rely on site-specific transcriptional pausing to regulate bacterial gene expression.

Transient RNAP pausing throughout transcription has been reported to be centrally involved in sequential RNA folding and in the recruitment of key regulatory factors[45]. RNAP pausing was also previously shown to be involved both in sensing environmental cues and controlling gene expression, as previously reported for the *pyrBI* leader[46], where uridine triphosphate (UTP)-specific transcriptional pausing controls translational coupling and intrinsic termination. Similarly, our results show that *thiC* transcriptional pausing is intimately linked to riboswitch TPP sensing and gene regulation, consistent with a recently predicted kinetic model[47]. On the basis of our study, we propose a mechanism in which TPP sensing is efficiently performed at pause sites A138 and C158, which are expected to provide more time for metabolite sensing (Fig. 6, left panel). Importantly, the formation of the anti-P1 stem at C187 suggests that transcriptional complexes located within a narrow transcriptional window between positions G100 and C187 efficiently perform TPP

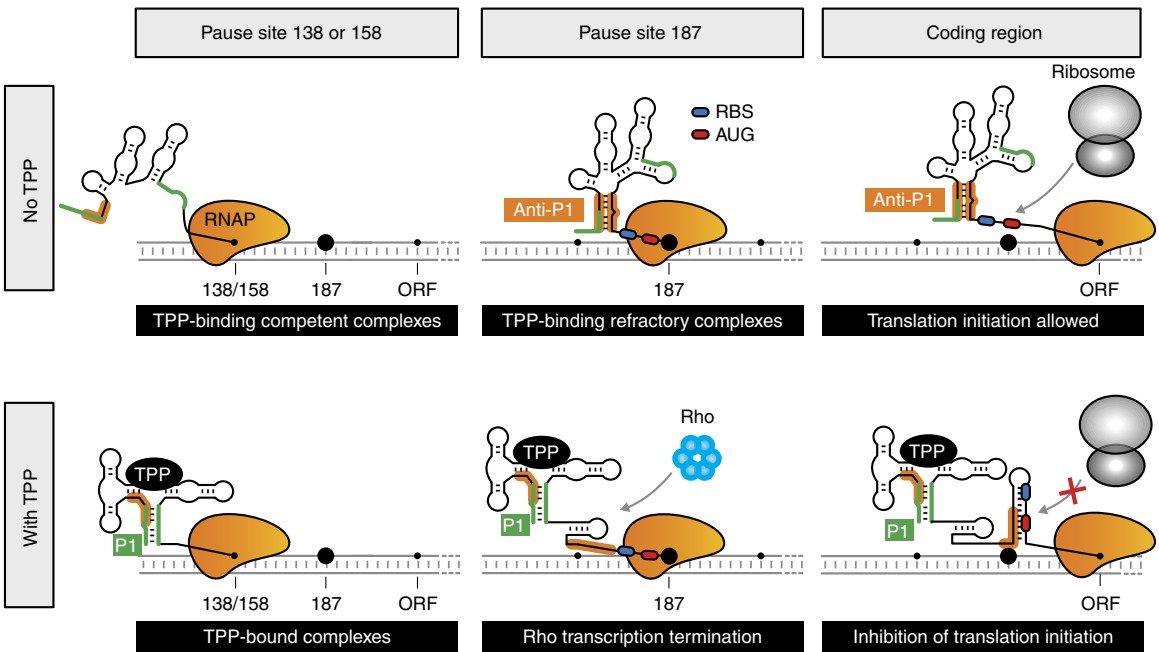

**Figure 6 | Transcriptional and translational regulatory mechanisms of the *thiC* riboswitch.** Upper panel: in absence of TPP, transcription elongation complexes located at pause sites 138 or 158 exhibit TPP-binding competent riboswitch structures. However, elongation complexes reaching pause site 187 adopt the anti-P1 structure that strongly reduces TPP-binding affinity. Elongation complexes reaching the coding region allow efficient 30S ribosomal subunit binding and translation initiation. Lower panel: in presence of TPP, elongation complexes located at pause sites 138 or 158 exhibit *thiC* riboswitch structures, in which the P1 stem is formed. On reaching the 187 pause site, a significant fraction of elongation complexes undergo Rho transcription termination. However, elongation complexes carrying on transcription into the coding region allow the formation of the sequestering stem, inhibiting translation initiation.

sensing. This corresponds to ∼1.6% of the *thiC* operon length, explaining the strong preference of the *thiC* riboswitch to bind ligand cotranscriptionally (Fig. 2d,f). The importance of pausing in the *thiC* RBS/AUG region is consistent with recent nascent elongating transcripts sequencing data showing a major pause site at position U188 *in vivo*[10]. Our study indicates that in the absence of TPP, formation of the anti-P1 stem at C187 locks the riboswitch in the ON state, which markedly reduces the metabolite-binding affinity of EC-187 (Fig. 6, middle panel). However, the majority of incoming TPP-bound transcription complexes reaching position 187 undergo Rho-dependent transcription termination (Fig. 6, middle panel). As long as transcription complexes clear the pause site, translation initiation is modulated accordingly (Fig. 6, right panel), providing a fail-safe mechanism at the translational level for TPP-bound complexes not terminated by Rho. Taken together, our results show that the C187 pause site acts as a central checkpoint where gene expression is either secured or repressed.

In *E. coli*, Rho-dependent termination is involved in many processes, including resolving RNA–DNA hybrids, and coupling transcription and translation through polarity[48,49]. Although polarity can result from inefficient ORF translation allowing Rho to bind to RNA, an additional polarity model has been described in which Rho and the ribosome act as alternative partners for the RNAP-bound NusG factor[48,50]. In this model, the formation of ribosome–NusG or Rho–NusG complex ensures either ribosome translation or transcription termination, respectively. The impact of NusG on *thiC* Rho-dependent termination (Fig. 1f) strongly suggests that a NusG-based mechanism, rather than a translation inhibition mechanism, directly controls *thiC* mRNA levels, as found in constitutive Rho-dependent terminators. Furthermore, in contrast to *E. coli ribB* and *Salmonella enterica mgtA* riboswitches[6], the *thiC* riboswitch modulates both *rut* site accessibility and pause half-life on metabolite binding, making

the *thiC* riboswitch the first example of a mechanism that controls Rho transcription termination through both processes. Our results indicate that the *thiC* riboswitch uses a novel mechanism that selectively modulates *rut* site accessibility through metabolite-induced RNA conformational changes to regulate transcription elongation (Supplementary Fig. 3c). This is in contrast to other models in which transcription is allowed to proceed by inactivating Rho either through Hfq[51] or Rho-antagonizing RNA elements[22]. Furthermore, our data indicate that Rho termination requires the TPP-mediated increase in C187 pause time, which is abolished by destabilizing the 160–175 nt hairpin structure (Fig. 3b, U160A mutant). Such a regulatory mechanism ensures direct control of transcription elongation through TPP-mediated *thiC* riboswitch conformational changes.

The importance of pause sites for riboswitch regulation was first established for the kinetically controlled *B. subtilis ribD* riboswitch[27], where RNAP pausing was found to prevent antiterminator formation. The *thiC* riboswitch represents a more sophisticated level of gene regulation since it selectively prevents and promotes anti-P1 stem formation at pause sites 158 and 187, respectively. Such a stepwise transcription process enables potent TPP sensing (EC-158) and accurate signal transduction (EC-187) to produce the required genetic outcome. Additional studies implied that transcriptional pausing could coordinate aptamer and expression platform folding[25,28]. Although the precise role of RNAP pausing on riboswitch folding is still to be determined in those cases, a discontinuous transcription mechanism is likely important for RNA folding and metabolite sensing. Even though additional experiments are required to ascertain the single nucleotide resolution mapping of transcriptional pause sites, our findings nevertheless clearly demonstrate the presence of RNAP pausing within the RBS/AUG region of *E. coli* riboswitches. Interestingly, such pauses were also detected *in vivo* for the *lysC* and *thiM*

riboswitches[10], consistent with transcriptional pausing being important for gene regulation. Further work will be required to ascertain the influence of transcriptional pausing within *E. coli* riboswitch sequences. The presence of multiple transcriptional pause regions might indicate strong evolutionary pressure on riboswitches to maintain specific metabolite-binding capabilities and transcriptional pausing activities. From an evolutionary standpoint, the presence of already existing pause sites allowing efficient translation initiation probably facilitated the establishment of Rho-dependent transcription termination mechanisms. The identification of pause sites in a wide range of bacterial transcripts strongly suggests that transcriptional dynamics is important for the regulation of bacterial mRNAs.

## Methods

**DNA oligonucleotides and bacterial strains.** DNA oligonucleotides were purchased from Integrated DNA Technologies. Strains used in this study were derived from *E. coli* MG1665 and are described in Supplementary Table 3. The strain DH5α was used for routine cloning procedures[52]. The strain BL21 (DE3) was used for overproduction of Rho, NusG and RNAP proteins. The *thiC* transcriptional and translational fusions were constructed using the PM1205 strain[5]. The PCR strategy to obtain *thiC* riboswitch translational constructs is described in Supplementary Table 4 and the oligonucleotides used are described in Supplementary Table 5. Transcriptional fusions required an additional PCR step (AD17 and oligo-stop oligonucleotides) to insert a stop codon. Mutations performed in *thiC* were made using a three-step PCR strategy (Supplementary Table 4). Briefly, PCR1 and PCR2 were performed using genomic DNA and PCR3 was performed using PCR1 and PCR2 products. The procedure to insert pfr-delta *thiC* in the *rne-131* strain was achieved by delivering a single copy of the *thiC* transcriptional fusion at the lambda *att* site. Stable lysogens were screened using PCR reactions[5,53]. All obtained *lacZ* fusions were sequenced to ensure the integrity of constructs.

**β-galactosidase assays.** Kinetic assays for β-galactosidase experiments were performed as described previously[5]. Briefly, an overnight bacterial culture grown in M63 minimal medium containing 0.2% glycerol was diluted to an $OD_{600}$ of 0.02 in 50 ml of fresh medium. The culture was incubated at 37 °C until an $OD_{600}$ of 0.1 was obtained. Arabinose (0.1%) was added to induce the expression of *lacZ* constructs. β-galactosidase experiments assessing the involvement of Rho were performed in 3 ml of culture media as described above. TPP (0.5 mg ml$^{-1}$) and bicyclomycin (25 μg ml$^{-1}$) were added as indicated. The experiments were performed in triplicate and the average values with standard deviations (s.d.) are reported. Descriptions of the mutants are in Supplementary Table 4.

**RNase H probing assays.** The PCR strategy to obtain DNA templates used for transcription reactions is described in Supplementary Table 6. Aliquots of transcription reactions (described in Methods) were mixed with 20 μM DNA oligonucleotides for 5 min. RNase H cleavage was initiated by adding a solution of RNase H in 5 mM Tris-HCl, pH 8.0, 20 mM MgCl$_2$, 100 mM KCl, 50 μM EDTA and 10 mM 2-mercaptoethanol at 37 °C for 5 min. Reactions were stopped by adding an equal volume of a stop solution (95% formamide, 20 mM EDTA and 0.4% SDS). The experiments were performed in triplicate and the average values with s.d.'s are reported. To monitor co- or post-transcriptional TPP binding, the transcription reaction was mixed with 200 μM DNA and RNase H for 15 s at 15 s, 45 s, 90 s, 2 min and 3 min after TPP addition. The reported errors for the TPP-binding rates are the s.e. in the fitting, which is assumed to be approximated by the standard deviation[54]. Images have been cropped for presentation. Full size images are presented in Supplementary Fig. 16.

**Native SHAPE probing.** The SHAPE reaction was essentially prepared as previously described[35]. Briefly, native transcription ECs were obtained by incubating 20 mM Tris-HCl, pH 8.0, 100 mM KCl, 20 mM MgCl$_2$, DNA template, *E. coli* RNA polymerase and sigma70 factor at 37 °C for 3 min. Reactions were performed in the absence or presence of 10 μM TPP. Transcription reactions were completed by adding NTP at 37 °C for 10 min. The EC-187 complex was obtained by transcribing a DNA template containing a biotin on the antisense strand 5′-end. Nascent RNA molecules were subjected either to 10% DMSO or 150 mM NAI for 15 min at 37 °C. Transcription reactions were quenched by passing complexes through G50 gel filtration columns and purified RNAs were collected in 0.5 × TE buffer. Selectively acylated RNA was incubated with different radioactively labelled DNA oligonucleotides to cover the complete riboswitch (1494FPJ, 1495FPJ, 1789FPJ and 1790FPJ) at 65 °C for 3 min, then cooled to 22 °C for 5 min. Reverse transcription reactions were performed at 55 °C for 15 min. Reactions were next quenched by incubating at 95 °C for 5 min. RNA molecules were hydrolysed using 200 mM NaOH, which was neutralized using a Tris-HCl-buffered solution[35]. Products were resolved by denaturing polyacrylamide gel electrophoresis and

visualized by phosphorimaging. Images have been cropped for presentation. Full size images are presented in Supplementary Fig. 16.

**Structural bioinformatics.** To predict the secondary structure of the *thiC* riboswitch in the absence of TPP, intergenic sequences located upstream of the *thiC* were retrieved in proteobacteria using the RiboGap database (http://ribogap.iaf. inrs.ca). Using these sequences, RNA secondary structure prediction was performed using both LocaRNA[55] and CMfinder[56] from which a potential candidate exhibiting an unpaired RBS region was obtained. The validity of the prediction was further tested by analysing the stability of local optimal structures using RNAConSLOpt[57]. Most predictions corresponded to OFF models, but the single predicted ON conformation was close to the LocaRNA and CMfinder models. The predicted secondary structure model for an alignment of 77 gammaproteobacteria (Supplementary Data) is represented in Supplementary Fig. 1a and shows the presence of an alternative pairing (anti-P1 stem) preventing both P1 and sequestrator helices (Supplementary Fig. 1b), consistent with a TPP-free ON state structure allowing efficient translation initiation. Using the ON state structure, an additional homology search was performed using INFERNAL on all intergenic regions with predicted TPP riboswitches in sequenced genomes[58] and resulting hits were realigned with cmalign to confirm the presence of the model[59]. A total of 127 additional sequences were found to contain the anti-P1 stem domain, but were not included due to additive minor divergences of the ON state riboswitch structure. In our model, the ON state structure of the *thiC* riboswitch (Supplementary Fig. 1b) is mutually exclusive to the TPP-bound conformer, thereby providing a molecular mechanism explaining how the *thiC* riboswitch modulates translation initiation on TPP binding.

**In vitro transcription.** The PCR strategy to obtain DNA templates used for transcription reactions is described in Supplementary Table 6. *In vitro* transcriptions were performed in 20 mM Tris-HCl pH 8.0, 20 mM MgCl$_2$, 20 mM NaCl, 14 mM 2-mercaptoethanol and 0.1 mM EDTA. The DNA template, sigma70 factor and RNAP were incubated at 37 °C for 5 min. The UAA trinucleotide, CTP/GTP nucleotides and [α-$^{32}$P] UTP were then added and the reaction incubated at 37 °C for 8 min, thus yielding an ECs stalled at position 14 (EC-14). The sample was passed through G50 columns to remove any free nucleotides. Transcription reactions were completed by adding all four nucleotides with heparin to allow only one round of transcription. TPP was added at 10 μM when indicated. Time pausing experiments were either performed using 25 or 100 μM NTPs. For time done at 25 μM NTP, time aliquots were taken from 10 s to 25 min. Time aliquots of 15 s, 30 s, 45 s, 60 s, 90 s, 2 min, 3 min, 4 min, 5 min, 8 min and 10 min were used for reactions done at 100 μM NTP. A chase reaction was done by adding 1 mM NTP. The half-life of transcriptional pausing was done by determining the fraction of each RNA species compared with the transcription readthrough for each time point, which was analysed with pseudo-first-order kinetics to extract the half-life[60]. For each determination, we have subtracted the background signal. Transcriptional sequencing were performed by including 3′-O-methyl-NTP in the reactions at varying concentrations to only permit a small fraction of transcriptional arrests[28]. Images have been cropped for presentation. Full size images are presented in Supplementary Fig. 16.

**Preparation of transcription ECs.** The description of PCR constructs used for the preparation of ECs can be found in Supplementary Table 6. Transcription were performed as described above using DNA templates containing a biotin at the 5′- end of the antisense strand. Streptavidin was added to a ratio of 2.5:1 with DNA and incubated for 5 min before adding NTP to complete the transcription reaction. Control experiments using cobalt beads were performed as described before[61]. Beads were resuspended in a solution of 95% formamide, 20 mM EDTA and 0.4% SDS after the wash step. Images have been cropped for presentation. Full size images are presented in Supplementary Fig. 16.

**Rho transcription termination assays.** *In vitro* transcriptions were performed as described above using a transcription buffer containing 40 mM Tris-HCl, pH 8.0, 50 mM KCl, 5 mM MgCl$_2$ and 1.5 mM DTT. Rho (50 nM) was added in the first step of the transcription reaction. Transcriptions were completed by adding 50 μM NTPs and rifampicin (20 μg ml$^{-1}$). The NusG factor (50 nM) was added where indicated. Resulting reactions were treated with a phenol/chloroform extraction and mixed with an equal volume of a stop solution containing 95% formamide, 10 mM EDTA and 0.1% SDS. Images have been cropped for presentation. Full size images are presented in Supplementary Fig. 16.

**In vitro translation.** The description of the PCR strategy to obtain DNA templates used for transcription–translation assays can be found in Supplementary Table 6. *In vitro* translations were performed with the PURExpress Kit from New England Biolabs. Reactions were performed in the presence of transcription and ribosome solutions, 0.1 μM *thiC* and *lacZ* DNA template, 0.16 μM *E. coli* RNA polymerase, 0.32 μM sigma70 factor and [$^{35}$S]-methionine at 37 °C for 2 h. Reactions were stopped by placing samples on ice for 10 min. Acetone 100% was added to a 4:1 ratio and the resulting solution incubated for 15 min at 4 °C. Precipitation was

performed by centrifugation at 15,900 g for 10 min. Pellets were resuspended in denaturing buffer. Samples were resolved on SDS–polyacrylamide gel electrophoresis and gels were visualized using a Typhoon FLA 9500 (GE Healthcare). For each experiment, the expression of ThiC was normalized to that of LacZ, which acts as an internal control not modulated by the presence of 25 μM TPP. Titrations were performed using TPP concentrations corresponding to 50 nM, 100 nM, 250 nM, 1 μM, 2.5 μM, 5 μM, 10 μM, 25 μM, 50 μM and 100 μM. For uncoupled transcription–translation assays, the reaction was initiated by performing a transcription step for 15 min at 37 °C using a transcription solution without amino acids/transfer RNA. Rifampicin (250 μg ml$^{-1}$) was next added to the reaction and incubated for 1 min to prevent transcription re-initiation. Translation was initiated by adding a solution containing amino acids and transfer RNA and was incubated for 15 min at 37 °C. Reactions were stopped as described for transcription–translation-coupled assays. Images have been cropped for presentation. Full size images are presented in Supplementary Fig. 16.

**Northern blot analysis.** The Northern blot analysis was performed as previously described[5]. Briefly, bacteria were grown at 37 °C in M63 0.2% glucose minimal medium to midlog phase. Total RNA was extracted using using hot phenol[62]. Rifampicin (250 μg ml$^{-1}$) was added to block transcription. The DNA probe used for the detection is AD113. Images have been cropped for presentation. Full size images are presented in Supplementary Fig. 16.

**Primer extension analysis.** The determination of the TSS was done as previously reported[28]. Briefly, 40 μg of total RNA was incubated using a radioactively labelled DNA oligonucleotide. Reverse transcription reactions were performed according to the manufacturer protocol and reactions were precipitated and resolved on denaturing gels. PCR reactions were used as sequencing markers. The DNA oligonucleotide used for the primer extension is AD5. Images have been cropped for presentation. Full size images are presented in Supplementary Fig. 16.

**Data availability.** The authors declare that all data supporting the findings of this study are available within the article and its Supplementary Information, or from the corresponding author on request.

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

## Acknowledgements

We thank members of the Lafontaine laboratory and Dr Carlos Penedo and Dr Alain Lavigueur for discussion. J.P. and D.A.L were supported by the Fonds de Recherche du Québec—Santé as Junior 1 and Senior Scholar, respectively. This work was supported by the Natural Sciences and Engineering Council of Canada (J.P.) and Canadian Institutes of Health Research (D.A.L.).

## Author contributions

A.C., F.P.-J., J.P. and D.A.L. planned and analysed the experiments. A.C., F.P.-J., J.-C.B.-D., L.B., M.R.N., A.D. and P.T. performed the experiments. A.C., F.P.-J., J.P. and D.A.L. wrote the paper.

## Additional information

**Competing financial interests:** The authors declare no competing financial interests.

