## [Peer Review File · Nature Communications]

Reviewers' comments:

Reviewer #1 (Remarks to the Author):

This is a very thorough paper stressing the importance of pause sites (normally sequence dependent) on the dynamics and mechanisms of riboswitching (in the present case of the TPP riboswitch).

That the effects of ligand on the switching is kinetically driven and occurring co-transcriptionally are accepted facts. The role of pause sites is less addressed. Thus, this paper is very timely.

Since such a mechanism would be also polymerase dependent, it would be interesting to know whether the authors have tried such experiments using a fast polymerase like the T7 polymerase.

These types of test when dealing with pause sites have been tried in *Genes Dev.* 2009 23: 2650-2662.

Reviewer #2 (Remarks to the Author):

The manuscript by Lafontaine and Perreault describes a detailed biochemical study on riboswitch mediated gene control. While the basic concept of riboswitches is easy to grasp, their mechanism of action is rather complex and has remained elusive. One of these aspects concerns the involvement of transcriptional pausing that impacts folding of the nascent RNA. Pause sites are likely responsible to enlarge the time window for ligand sensing of folding intermediates, and hence, are assumed to represent a crossroad into different folding pathways.

In this respect, the team investigated the molecular mechanism of the *E. coli* thiC TPP riboswitch. The authors put an enormous amount of work into their study. They used three complementary methods (β galactosidase assays, RNase H probing, SHAPE) to deliver a quite consistent model of how this riboswitch transcription and translation are timed to regulate gene expression, or more precisely how eg different positions of the transcription elongation complex at the RNA impact the interdependent secondary structure alternatives (P1 stem stabilization versus anti P1 stem formation) and how these structures affect TPP binding/affinity; and moreover, how eg Rho/NusG affects transcription performance/termination of this particular riboswitch.

In my opinion, all experiments seem properly performed with the necessary controls and repetitions. Furthermore, the conclusions made are convincing and shed much-needed light on the complex mechanism of riboswitches. I also want to state that studies of this kind are very important to eventually accelerate progress towards the application of riboswitches as drug targets.

I very much support publication without the need for further experimentation.

There are very minor spelling errors and format inconsistencies throughout the manuscript and in the reference list (eg thiamin; doi is sometimes given, sometimes not, etc.)

Reviewer #3 (Remarks to the Author):

Chauvier et al. report a landmark study of the role of transcriptional pausing in regulation of the thiC riboswitch that provides a new model for the role of translation start-site pauses. This work is more nuanced than prior studies of pausing in riboswitch control regions in that it defines different roles for different pause sites (although it is consistent with the prior studies). More generally, the work builds on a very long history defining the roles of transcriptional pausing in attenuation control mechanisms, and is, without question, a major new contribution to this important field. (The field is important not just because transcriptional pausing is a vital component of bacterial gene regulation, but also it because is a crucial contributor to regulation of gene expression in metazoans, including during human development). Chauvier et al. show that there are three major pause sites in the thiC leader region, but that they have a different consequences for regulation. The first two pauses, at ~138 and ~158 nt, provide time windows for thiamine pyrophosphate (TPP) binding to the riboswitch RNA, which stabilizes an RNA conformation that blocks translation and is targeted by Rho for termination. In contrast, the third pause, at the AUG start codon for thiC, provides a time window for rearrangement of the leader RNA into a structure that cannot bind TPP and that subsequently promotes efficient translation of thiC. The authors establish this model with multiple types of experimental evidence, including in vivo gene expression, in vitro transcription, in vitro translation, oligo binding/RnaseH-cleavage experiments, and RNA SHAPE analysis. These experiments are expertly performed and, with one exception, provide compelling evidence for the authors interpretation and models. Thus, I strongly support publication of a revised form of this manuscript. It will have major impact.

However, there are two overarching issues, one stylistic and one substantive, as well as numerous minor points that must be dealt with before the manuscript can be published. Fortunately, the overarching issues do not fundamentally affect the authors' conclusions about gene regulation and the model they propose (and they are easily addressed). These issues are enumerated below.

Major issues

1. The authors have a tendency to overstate the novelty of their findings. They may feel this is a requirement to secure high-profile publication, but it actually detracts from the impact of the work because it gives the reader an impression the authors are over-hyping their findings rather than letting the results stand on their own merit, which is substantial. For example (p4 li23), they authors write that they describe the first molecular mechanism in which a transcriptional pause site is involved both in sensing environmental cues and controlling gene expression. That claim is simply untrue. To give one example, pause sites in the pyrBI leader region sense UTP concentration and control gene expression via translational coupling and intrinsic termination (Donahue and Turnbough, 1994 JBC 269: 18185). There are many others, worked out to varying degrees. Another example of this tendency occurs in the discussion (p23 li10) where the authors assert that no other study has found pausing involved in both positive and negative control. I'm unsure what that even means, since positive or negative regulation depends point of reference, but the claim adds nothing to the manuscript other than an impression the authors are overhyping their results. The paragraph goes on to make unsupported claims about pause site mapping (see major point 2), which furthers

this impression. The authors have provided the first validation of the start-codon pausing hypothesis and an elegant new model. Rather than claiming great novelty or to be the first to show something, the authors would be better served to simply describe their results and their model clearly. Many journals have an editorial policy against claims of priority, in part for this reason. I'm unsure of Nature Communications policy, but the authors would be best served by avoiding such claims.

2. The authors claim to map transcription pause sites to locations that do not match those predicated by a previously described paused consensus sequence. However, the evidence they provide for this claim is either unconvincing or not shown at all. In short, I am not persuaded that the authors have mapped the pause sites shown in Fig. 5a with single-nt accuracy. The authors either must modify the manuscript to indicate they have not mapped pauses to single-nt resolution or present convincing evidence of single-nt resolution mapping. As best I can tell, the only data the authors provide to justify their pause site assignments is for the three thiC pauses (Fig. S6). No comparable data are provided for the thiB, thiM, lysC, ribB, or btuB pauses.

In the experiment shown in Fig. S6, the authors use 3'-O-methyl NTPs as chain terminators to generate an RNA sequence ladder, and then electrophorese this ladder alongside the pause RNAs from an in vitro transcription experiment. There are several problems with this mapping experiment. First, resolving RNAs to single-nt resolution in the 100-200 nt range by electrophoresis is hard. Some RNA structure persists during electrophoresis in denaturing gels, leading to irregular band spacing and to compressions. Second, RNAs containing a 3' O methyl group do not co-migrate exactly with their cognate 3'OH RNAs. The 3' O-methyl and 3'-OH RNAs are chemically different and thus migrate slightly differently. Depending on the RNA, the spacing, and possible compressions, this can make precise alignment of the sequence ladder with a non-methylated RNA nearly impossible. Third, the authors do not run the ladder directly next to the pause bands they are trying to assign! Guessing at the alignment while looking across many lanes is problematic, and I cannot make a position assignment from the data in Fig. S6. Fourth, the authors do not provide a sequence for the bands shown in the sequence ladder by labeling the bands beside the gel. Such labeling is a minimum requirement for presenting sequence mapping data. The absence of band assignments makes it very difficult for a reader to assess the authors' results. The right way to do this experiment (at a minimum) is to run the pause RNA directly next to the ladder and also, in a different set of lanes, to mix the pause RNA with the ladder samples. The latter step (mixing the RNA to be mapped with the sequence ladder samples) is necessary to ensure that differences in salt concentration etc. do not alter the mobility of bands and cause mis-assignments. All of these steps need to be taken and presented in a figure that unambiguously documents the pause position assignments before a claim of mapping pauses to unexpected positions can be supported. The maxim "extraordinary claims require extraordinary evidence" applies here.

There are additional steps the authors can (and should) take to test their pause location assignments. First, have the authors examined the pause data published in ref. 10? It should be available for download from NCBI GEO. If the thiC pauses (and other pauses) occur at non-canonical positions, then it should be possible to find the non-canonical pauses in the published data. If the pause positions differ in the published data, then it would raise a question of whether the pauses in vivo occur at positions that differ from those observed by the authors in vitro. Second, the authors can test the effect of limiting NTP concentrations on the dwell time of RNA polymerase at the pause sites. Pause lifetimes always are sensitive to the concentration of the incoming NTP. Thus, if the "C187" pause truly occurs before addition of UTP, lowering the concentration of UTP should increase

the pause dwell time and lowering the concentration of ATP, GTP, or CTP should not increase the pause dwell time comparably (e.g., 10 uM UTP, 100 uM ATP, CTP, GTP vs. 10 uM GTP, 100 uM ATP, CTP, UTP).

Finally, where are the mapping data for the thiB, thiM, lysC, ribB, or btuB pauses? All these data will need to be presented as described above (as supplemental figures) before the authors would be justified in publishing claims of that the pauses occur at non-canonical positions.

Minor issues

1. p3 li3 “regulate” is a transitive verb and needs an object, ie, “regulate gene expression”.
2. p11 li4, Although the authors describe conducting translation assays uncoupled from transcription in the methods, the description here is of a coupled assay (li 1). Thus, it is unclear how the authors can conclude the 2-fold reduction is an effect on translation and not an effect on transcription (RNA level). The paragraph should be reworded to make it clear how this conclusion can be reached.
3. p13 li 4 and Fig. S5. The fast rates (0.2 s⁻¹) described here are actually indeterminate because the reaction is over at the first time point. The statement instead should be that the fast binding rate has a lower bound of 0.2 s⁻¹.
4. The statement that the last 12 transcribed residues are contained in the RNAP exit channel is confusing. Ref. 33 reports that 14 nt of RNA are protected within RNAP. Of these, 9-10 nt are in the RNA;DNA hybrid and 4-5 nt are in the RNA exit channel.
5. Supplementary Figure 3D is not really convincing. The effects upon addition of Rho are minor. Is this assay done in presence of TPP? If not, why is C137 A138 well accessible in the -Rho lane? Why are the other reverse changes in L5, L3A and C115 (observed +TPP) not observed upon addition of Rho?
6. p14 li 15: “U160-A175 hairpin”, do the authors mean the G149-C187 hairpin, or perhaps local RNA structure in the upper half of this hairpin?
7. p15 li 1,2 How was mutation U186A identified? If the pause is at C187, was a mutation in the C tested. How do the authors explain the effect of this mutation, since the RNA structure is not affected apparently?

8. p18, li3 the correct abbreviation of SHAPE is selective 2' hydroxyl acylation analyzed by primer extension.

9. How do the authors explain that nt 7-17 in the P1 stem of the full length RNA ON state (sup. Fig. 10c) is highly reactive in SHAPE RNA structure probing while this is predicted to be base paired? In fact, the largest change in SHAPE reactivity is found in nt 68-73 (p5 loop; sup fig. 10d).

10. There is no major difference base pairing of nucleotides 10-19 (probe P1) between the OFF state (fig. 1a) and ON state (sup Fig. 1B). In both the structures nt 10-15 is base paired and nt 16-19 is single stranded. How then is RNaseH probing with P1 informative? Are the positions of this probe correct. I cannot find the sequence in material and methods.

All corrections performed in the manuscript are highlighted using the “Track changes” function in Word.

Reviewer #1.

This is a very thorough paper stressing the importance of pause sites (normally sequence dependent) on the dynamics and mechanisms of riboswitching (in the present case of the TPP riboswitch). That the effects of ligand on the switching is kinetically driven and occurring co-transcriptionally are accepted facts. The role of pause sites is less addressed. Thus, this paper is very timely. Since such a mechanism would be also polymerase dependent, it would be interesting to know whether the authors have tried such experiments using a fast polymerase like the T7 polymerase. These types of test when dealing with pause sites have been tried in *Genes Dev.* 2009 23: 2650-2662.

Response by Authors to Reviewer #1.

We thank reviewer #1 for the positive comments about our study. We agree that while the ligand effect has been previously characterized, much remains to be explored about the role of pause sites on riboswitch regulation. As mentioned by the reviewer, the negative impact of using the T7 RNA polymerase on riboswitch activity has been described in the case of the pH-responsive riboregulator (Nechooshtan et al, *Genes Dev.* 2009). Interestingly, it has also been demonstrated for the AdoCbl-dependent riboswitch (Perdrizet et al, *PNAS*, 2012) that the use of a T7-made riboswitch transcript yielded incorrect folding. Given those clear precedents, we do not foresee the necessity to add such experiments in our study. Since our work is focused on the role of the C187 pause region on riboswitch regulation and how it affects TPP sensing, we consider that our analysis on the thiC riboswitch pause sites is sufficient on its own, which provides a novel concept about riboswitch regulation.

Reviewer #2.

The manuscript by Lafontaine and Perreault describes a detailed biochemical study on riboswitch mediated gene control. While the basic concept of riboswitches is easy to grasp, their mechanism of action is rather complex and has remained elusive. One of these aspects concerns the involvement of transcriptional pausing that impacts folding of the nascent RNA. Pause sites are likely responsible to enlarge the time window for ligand sensing of folding intermediates, and hence, are assumed to represent a crossroad into different folding pathways.

In this respect, the team investigated the molecular mechanism of the *E. coli* thiC TPP riboswitch. The authors put an enormous amount of work into their study. They used three complementary methods (β galactosidase assays, RNase H probing, SHAPE) to deliver a quite consistent model of how this riboswitch transcription and translation are timed to regulate gene expression, or more precisely how eg different positions of the transcription elongation complex at the RNA impact the interdependent secondary structure alternatives (P1 stem stabilization versus anti P1 stem formation) and how these structures affect TPP binding/affinity; and moreover, how eg Rho/NusG affects transcription performance/termination of this particular riboswitch.

In my opinion, all experiments seem properly performed with the necessary controls and repetitions. Furthermore, the conclusions made are convincing and shed much-needed

light on the complex mechanism of riboswitches. I also want to state that studies of this kind are very important to eventually accelerate progress towards the application of riboswitches as drug targets. I very much support publication without the need for further experimentation.

There are very minor spelling errors and format inconsistencies throughout the manuscript and in the reference list (eg thiamin; doi is sometimes given, sometimes not, etc.)

Response by Authors to Reviewer #2.

We thank reviewer #2 for the positive comments about our study. We have proofread the manuscript and ensured that the references are presented in a consistent manner.

Reviewer #3.

Chauvier et al. report a landmark study of the role of transcriptional pausing in regulation of the thiC riboswitch that provides a new model for the role of translation start-site pauses. This work is more nuanced than prior studies of pausing in riboswitch control regions in that it defines different roles for different pause sites (although it is consistent with the prior studies). More generally, the work builds on a very long history defining the roles of transcriptional pausing in attenuation control mechanisms, and is, without question, a major new contribution to this important field. (The field is important not just because transcriptional pausing is a vital component of bacterial gene regulation, but also it because is a crucial contributor to regulation of gene expression in metazoans, including during human development). Chauvier et al. show that there are three major pause sites in the thiC leader region, but that they have a different consequences for regulation. The first two pauses, at ~138 and ~158 nt, provide time windows for thiamine pyrophosphate (TPP) binding to the riboswitch RNA, which stabilizes an RNA conformation that blocks translation and is targeted by Rho for termination. In contrast, the third pause, at the AUG start codon for thiC, provides a time window for rearrangement of the leader RNA into a structure that cannot bind TPP and that subsequently promotes efficient translation of thiC. The authors establish this model with multiple types of experimental evidence, including in vivo gene expression, in vitro transcription, in vitro translation, oligo binding/RnaseH-cleavage experiments, and RNA SHAPE analysis. These experiments are expertly performed and, with one exception, provide compelling evidence for the authors interpretation and models. Thus, I strongly support publication of a revised form of this manuscript. It will have major impact.

However, there are two overarching issues, one stylistic and one substantive, as well as numerous minor points that must be dealt with before the manuscript can be published. Fortunately, the overarching issues do not fundamentally affect the authors' conclusions about gene regulation and the model they propose (and they are easily addressed). These issues are enumerated below.

Major issue #1. The authors have a tendency to overstate the novelty of their findings. They may feel this is a requirement to secure high-profile publication, but it actually detracts from the impact of the work because it gives the reader an impression the authors

are over-hyping their findings rather than letting the results stand on their own merit, which is substantial. For example (p4 li23), they authors write that they describe the first molecular mechanism in which a transcriptional pause site is involved both in sensing environmental cues and controlling gene expression. That claim is simply untrue. To give one example, pause sites in the pyrBI leader region sense UTP concentration and control gene expression via translational coupling and intrinsic termination (Donahue and Turnbough, 1994 JBC 269: 18185). There are many others, worked out to varying degrees. Another example of this tendency occurs in the discussion (p23 li10) where the authors assert that no other study has found pausing involved in both positive and negative control. I'm unsure what that even means, since positive or negative regulation depends point of reference, but the claim adds nothing to the manuscript other than an impression the authors are overhyping their results. The paragraph goes on to make unsupported claims about pause site mapping (see major point 2), which furthers this impression. The authors have provided the first validation of the start-codon pausing hypothesis and an elegant new model. Rather than claiming great novelty or to be the first to show something, the authors would be better served to simply describe their results and their model clearly. Many journals have an editorial policy against claims of priority, in part for this reason. I'm unsure of Nature Communications policy, but the authors would be best served by avoiding such claims.

Response by Authors to Major issue #1.

We thank reviewer #3 for the positive comments about our study. Overall, we took great care to reply to these issues and are confident that the revised version of our manuscript is significantly improved by addressing these comments. As suggested by the reviewer, we have rewritten parts of the manuscript to remove or edit sentences overstating our findings. However, while we agree with the reviewer that Donahue and Turnbough (JBC 1994) described a mechanism very similar to the one presented in our manuscript, we argue that it is important to differentiate our findings from other riboswitch studies. Thus, to avoid any confusion for the readers, we have modified two parts of our manuscript to more clearly reflect our thoughts about this point. We first changed the sentence at the end of the Introduction (p4 li23) for the following (underline indicate changes): "We describe the first riboswitch-based molecular mechanism in which a transcriptional pause site is involved in both sensing environmental cues and controlling gene expression." Next, to clearly state that such a relationship was previously demonstrated in other systems, we added a description of the pyrBI leader in the Discussion (p21 li8): "RNAP pausing was also previously shown to be involved both in sensing environmental cues and controlling gene expression, as previously reported for the pyrBI leader (Donahue and Turnbull, 1994), where UTP-specific transcriptional pausing controls translational coupling and intrinsic termination." With these two clarifications, we think that our findings are better contrasted with the current literature.

As suggested by the reviewer, we have removed the sentence "No other study has reported a transcriptional pause site involved in both positive and negative control of riboswitch-regulated genes."

Major issue #2. The authors claim to map transcription pause sites to locations that do not match those predicated by a previously described paused consensus sequence. However, the evidence they provide for this claim is either unconvincing or not shown at all. In short, I am not persuaded that the authors have mapped the pause sites shown in Fig. 5a with single-nt accuracy. The authors either must modify the manuscript to indicate they have not mapped pauses to single-nt resolution or present convincing evidence of single-nt resolution mapping. As best I can tell, the only data the authors provide to justify their pause site assignments is for the three thiC pauses (Fig. S6). No comparable data are provided for the thiB, thiM, lysC, ribB, or btuB pauses.

In the experiment shown in Fig. S6, the authors use 3'-O-methyl NTPs as chain terminators to generate an RNA sequence ladder, and then electrophorese this ladder alongside the pause RNAs from an in vitro transcription experiment. There are several problems with this mapping experiment. First, resolving RNAs to single-nt resolution in the 100-200 nt range by electrophoresis is hard. Some RNA structure persists during electrophoresis in denaturing gels, leading to irregular band spacing and to compressions. Second, RNAs containing a 3' O methyl group do not co-migrate exactly with their cognate 3'-OH RNAs. The 3' O-methyl and 3'-OH RNAs are chemically different and thus migrate slightly differently. Depending on the RNA, the spacing, and possible compressions, this can make precise alignment of the sequence ladder with a non-methylated RNA nearly impossible. Third, the authors do not run the ladder directly next to the pause bands they are trying to assign! Guessing at the alignment while looking across many lanes is problematic, and I cannot make a position assignment from the data in Fig. S6. Fourth, the authors do not provide a sequence for the bands shown in the sequence ladder by labeling the bands beside the gel. Such labeling is a minimum requirement for presenting sequence mapping data. The absence of band assignments makes it very difficult for a reader to assess the authors' results. The right way to do this experiment (at a minimum) is to run the pause RNA directly next to the ladder and also, in a different set of lanes, to mix the pause RNA with the ladder samples. The latter step (mixing the RNA to be mapped with the sequence ladder samples) is necessary to ensure that differences in salt concentration etc. do not alter the mobility of bands and cause mis-assignments. All of these steps need to be taken and presented in a figure that unambiguously documents the pause position assignments before a claim of mapping pauses to unexpected positions can be supported. The maxim "extraordinary claims require extraordinary evidence" applies here.

There are additional steps the authors can (and should) take to test their pause location assignments. First, have the authors examined the pause data published in ref. 10? It should be available for download from NCBI GEO. If the thiC pauses (and other pauses) occur at non-canonical positions, then it should be possible to find the non-canonical pauses in the published data. If the pause positions differ in the published data, then it would raise a question of whether the pauses in vivo occur at positions that differ from those observed by the authors in vitro. Second, the authors can test the effect of limiting NTP concentrations on the dwell time of RNA polymerase at the pause sites. Pause lifetimes always are sensitive to the concentration of the incoming NTP. Thus, if the "C187" pause truly occurs before addition of UTP, lowering the concentration of UTP

should increase the pause dwell time and lowering the concentration of ATP, GTP, or CTP should not increase the pause dwell time comparably (e.g., 10 uM UTP, 100 uM ATP, CTP, GTP vs. 10 uM GTP, 100 uM ATP, CTP, UTP).

Finally, where are the mapping data for the *thiB*, *thiM*, *lysC*, *ribB*, or *btuB* pauses? All these data will need to be presented as described above (as supplemental figures) before the authors would be justified in publishing claims of that the pauses occur at non-canonical positions.

Response by Authors to Major issue #2.

As mentioned by the reviewer, we agree that determining transcriptional pause sites with single nucleotide resolution is hard using 3'-O-methyl NTPs, mostly due to technical or experimental difficulties. However, as indicated by the reviewer, the single-nt resolution mapping of transcriptional pause sites is not crucial for the conclusion of our paper, which indicates that pausing in the RBS/AUG vicinity is very important for riboswitch metabolite sensing. Thus, instead of obtaining single-nt resolution data for our study, we followed the advice of the reviewer and modified the manuscript accordingly (see below) to indicate that we have not mapped pausing sites to single-nt resolution but rather to a narrow region. Furthermore, when the migration resolution allowed to discriminate at the nucleotide level (e.g., *thiM* and *thiB*), we experimentally observed multiple bands occurring in discrete regions of riboswitch sequences, thus indicating that the pausing occurs at multiple positions within discrete regions. We thus think that describing transcriptional pausing as regions is actually more relevant to our findings. Thus, taking into account the fact that we do not provide single-nt resolution mapping data and that transcriptional pausing may occur as discrete regions, we rephrased the text of our manuscript in several sections. We first rephrased the section (p14 li9) describing *thiC* transcriptional pause sites (underline indicate changes): "Three pause regions were observed in the riboswitch expression platform that encompassed positions A138, C158 and C187". Similarly, we rephrased the part (p19 li10) about transcriptional pause sites found in other *E. coli* riboswitches: "In all cases, we found a clear pausing region in the vicinity of the RBS and start codon (Fig. 5a)." The legend of the Figure 5a (p33 li19) was changed to clearly indicate to the reader that our results refer to transcriptional pausing regions and not to a single nucleotide attribution to pause sites (underline indicate changes): "Transcriptional assays indicate that pausing regions are found nearby each riboswitch start codon. A single position was used to designate each pausing region, centered around positions 187 (*thiC*), 136 (*thiB*), 152 (*thiM*), 309 (*lysC*), 249 (*ribB*) and 235 (*btuB*)." Lastly, we added the following sentences about the difficulty to perform single nucleotide resolution mapping of transcriptional pause sites in the Discussion (p23 li21): "Even though additional experiments are required to ascertain the single nucleotide resolution mapping of transcriptional pause sites, our findings nevertheless clearly demonstrate the presence of RNAP pausing within the RBS/AUG region of *E. coli* riboswitches." Consequently, we consider that it is now clear for the reader that we do not claim in our paper that we have performed a single-nt resolution mapping of transcriptional pause sites.

As suggested by the reviewer, we have added additional data showing the mapping of pause regions for the *thiB*, *thiM*, *lysC*, *ribB* and *btuB* riboswitches. Those data are included in the Supplementary Figure corresponding to each characterized riboswitch. In a few cases, the figures are derived from new experiments. During the process, we assigned new pausing regions (differing only by a few nucleotides with the previously determined ones) for the *lysC* and *ribB* riboswitches. Importantly, these new regions are still consistent with our conclusions. The Figure 5a and Supplementary Figures 13 and 15 have been edited accordingly. As mentioned by the reviewer, we agree that it is beneficial for the reader to have access to these new mapping data to note the extent of transcriptional pausing regions.

A direct consequence to the fact that we now describe transcriptional pausing as regions is that we have now removed any claims regarding whether transcriptional pausing detected in our study fall within or not the previously determined consensus pause sequence. For example, such claims for the C187 pause site that were found in the Introduction (p4 li12) and in the Results (p19 li8) sections have been removed. The other claim about *E. coli* riboswitches differing from the consensus pause sequence has also been removed (p19 li11). In fact, as mentioned by the reviewer, the main goal of our manuscript is to describe a mechanistic function for the transcriptional pausing occurring in the vicinity of the RBS/AUG region. We therefore think that the comparison of our pause sites to the consensus pause sequence is not critical for the riboswitch mechanism of gene regulation to be supported.

Lastly, as suggested by the reviewer, we have examined the data from NCBI GEO and are grateful for this as it enabled us to compare our data with riboswitch pause sites detected *in vivo*. For example, the NET-seq data corroborate the presence of a transcriptional pause in the *thiC* RBS/AUG region at position U188, emphasizing the importance of pausing in this region *in vivo*. As indicated above, since we now report transcriptional pausing as regions, we do not think that it is relevant to discuss in great details about a possible difference between NET-seq results and our *in vitro* data. As a result, we have added this sentence in the Discussion (p21 li19): “The importance of pausing in the *thiC* RBS/AUG region is consistent with recent NET-seq data showing a major pause site at position U188 *in vivo* (Larson et al, Science, 2014)”. Furthermore, transcriptional pausing for *thiM* and *lysC* riboswitches were also detected in the RBS/AUG region by the NET-seq study, consistent with our *in vitro* data. In the Discussion, we therefore rephrased the text (p24 li1) describing our findings about *E. coli* riboswitches (underline show changes): “Interestingly, such pauses were also detected *in vivo* for the *lysC* and *thiM* riboswitches (Larson et al, Science, 2014), consistent with transcriptional pausing being important for gene regulation.” We note that no *in vivo* pause site was found for the *btuB*, *ribB* and *thiB* riboswitches. However, we do not discuss this in our manuscript as it would go beyond the scope of our study.

Minor issues

Issue #1. p3 li3 “regulate” is a transitive verb and needs an object, ie, “regulate gene expression”.

Response: The sentence has been corrected as suggested by the Reviewer.

Issue #2. p11 li4, Although the authors describe conducting translation assays uncoupled from transcription in the methods, the description here is of a coupled assay (li 1). Thus, it is unclear how the authors can conclude the 2-fold reduction is an effect on translation and not an effect on transcription (RNA level). The paragraph should be reworded to make it clear how this conclusion can be reached.

Response: In the Supplementary Methods (“In Vitro Translation” section), the reviewer will find the description for both coupled and uncoupled transcription-translation assays. These assays were performed with the PURExpress system from NEB which is a cell-free system reconstituted from purified components necessary for *E. coli* translation. The system contains His-tagged versions of all protein factors, which greatly facilitates the purification of required proteins present in the PURExpress assay (Shimizu et al, Methods, 2005). The system was previously used in numerous cases to demonstrate translation-dependent gene regulation. For example, this assay was used to show that CsrA represses *pnp* at the translational level (Park et al, J Bac, 2015). To clarify this assay, we have modified the following sentence (p10 li22)(underline indicates changes): “To directly monitor TPP-dependent control of *thiC* translation initiation, we carried out transcription-translation-coupled *in vitro* assays using the cell-free PURExpress system (Shimizu et al, 2005).”

Issue #3. p13 li 4 and Fig. S5. The fast rates (0.2 s^{-1}) described here are actually indeterminant because the reaction is over at the first time point. The statement instead should be that the fast binding rate has a lower bound of 0.2 s^{-1} .

Response: As suggested by the reviewer, we have changed the sentence (p13 li4) describing the TPP binding rates (underline indicates changes): “Fitting analysis yielded a fast binding rate having a lower bound of $0.20 \pm 0.04 \text{ s}^{-1}$ for co-transcriptional TPP binding and an apparent binding rate of $0.05 \pm 0.01 \text{ s}^{-1}$ for post-transcriptional binding (Supplementary Fig. 5a), consistent with co-transcriptional ligand binding being ~4-fold more efficient.”

Issue #4. The statement that the last 12 transcribed residues are contained in the RNAP exit channel is confusing. Ref. 33 reports that 14 nt of RNA are protected within RNAP. Of these, 9-10 nt are in the RNA;DNA hybrid and 4-5 nt are in the RNA exit channel.

Response: As suggested by the reviewer, we have indicated that 14 nt are protected within RNAP (p17 li5)(underline indicates changes): “Indeed, given that the last 14 transcribed residues (positions 176-187) are protected within RNAP, ...”. We have also redone Figure 4d and Supplementary Figures 11-15 to account for this change.

Issue #5. Supplementary Figure 3D is not really convincing. The effects upon addition of Rho are minor. Is this assay done in presence of TPP? If not, why is C137 A138 well accessible in the –Rho lane? Why are the other reverse changes in L5, L3A and C115 (observed +TPP) not observed upon addition of Rho?

Response: In the Supplementary Figure 3d, nuclease S1 digestions were all performed in the presence of $10 \mu\text{M}$ TPP. This is why C137-A138 and all reported changes are

obtained in the absence of Rho given that the riboswitch is bound to TPP. While we agree that the effect of Rho binding on the RNA are minor, we have observed them across various experimental trials and are thus confident that they are small but meaningful.

Issue #6. p14 li 15: “U160-A175 hairpin”, do the authors mean the G149-C187 hairpin, or perhaps local RNA structure in the upper half of this hairpin?

Response: The mention “U160-A175 hairpin” effectively refers to the upper half of the hairpin. The complete hairpin stem is otherwise named “RBS sequestering stem” throughout the manuscript.

Issue #7. p15 li 1,2 How was mutation U186A identified? If the pause is at C187, was a mutation is the C tested. How do the authors explain the effect of this mutation, since the RNA structure is not affected apparently?

Response: The U186A mutant was identified through a mutational analysis that was based on transcription kinetics. Although various mutations were observed to alter C187 pausing, only U186A was found to allow the riboswitch to bind TPP relatively efficiently and to regulate translation initiation. These latter characteristics are important since they indicate that the sequence change does not significantly alter riboswitch function. In the case of U186A, the molecular mechanism altering transcriptional pausing was not addressed in the current manuscript as it falls outside of the scope of our study. As a provisional explanation, it is possible that the U186A mutant results in a different structure of the RNA-DNA hybrid, which was previously shown to affect transcriptional pausing (Bochkareva et al, EMBO J, 2012). Alternatively, the mutated sequence may transiently compete with the formation of the U160-A175 hairpin within the RNA exit channel, which could alter RNAP pausing. We are currently addressing the mechanistic details of the U186A mutation and expect to publish our findings in a follow up story.

Issue #8. p18, li3 the correct abbreviation of SHAPE is selective 2' hydroxyl acylation analyzed by primer extension.

Response: As suggested by the reviewer, we have corrected the SHAPE abbreviation in the text.

Issue #9. How do the authors explain that nt 7-17 in the P1 stem of the full length RNA ON state (sup. Fig. 10c) is highly reactive in SHAPE RNA structure probing while this is predicted to be base paired? In fact, the largest change in SHAPE reactivity is found in nt 68-73 (p5 loop; sup fig. 10d).

Response: As indicated by the reviewer, our SHAPE assays revealed that the region 7-17 nt in the context of the full length RNA—but not in EC-187—is reactive toward SHAPE in the absence of TPP (Fig. 4f). We attribute this result to the anti-P1 stem of the full length RNA showing a higher degree of thermal motion (structural flexibility or base pair "breathing") than in EC-187. This is mostly due to the presence of the 155-187 nt downstream sequence in the full length RNA allowing a structural exchange between the anti-P1 stem and the RBS sequestering stem. In our manuscript, we describe this structural feature with the following (p18 li15): "This is presumably due to an equilibrium shift toward the formation of the RBS sequestering stem that can form in the context of the full-length riboswitch, but not in the context of EC-187." Importantly, our SHAPE

data is consistent with EC-187 promoting the formation of the anti-P1 stem, which is further supported through mutational analysis and TPP reduced affinity in EC-187. Lastly, as mentioned by the reviewer, the Supplementary Figure 10d shows a large change for the P5 loop region in the presence of TPP (both for the full length and EC-187). The P5 loop was previously shown to form an interaction with the P3 stem upon metabolite binding in the context of the thiM riboswitch (Steen et al, Nature Protocols, 2011). These results indicate that both sequences are able to bind TPP when in large excess, which is consistent with our study.

Issue #10. There is no major difference base pairing of nucleotides 10-19 (probe P1) between the OFF state (fig. 1a) and ON state (sup Fig. 1B). In both the structures nt 10-15 is base paired and nt 16-19 is single stranded. How then is RNaseH probing with P1 informative? Are the positions of this probe correct. I cannot find the sequence in material and methods.

Response: RNase H assays rely on the hybridization of a DNA oligonucleotide to an RNA sequence whereby the inherent stability of the RNA structure plays a major role in the yield of the RNase H cleavage products. As mentioned in the **minor issue #9**, RNA structures exhibit varying degrees of inherent structural flexibility. Thus, given that our results show that the P1 probe (targeting region 10-19 nt) yields efficient RNase H cleavage activity in the absence of TPP, it suggests that this region is flexible and that the P1 probe can hybridize to this region. These results are supported by SHAPE assays also showing that this region is flexible in the full length RNA in the absence of TPP. In contrast, TPP binding to the riboswitch results in a very strong protection towards RNase H cleavage, suggesting that the TPP-aptamer complex is highly stable and is not bound by the P1 probe. Interestingly, similar results were obtained for the btuB riboswitch when using probes targeting the riboswitch core region (probes 169, 181 and 201, Perdrizet et al, PNAS 2012), where base paired regions are cleaved in the absence of ligand. Thus, in addition to the detection of the structural state of a given RNA molecule, RNase H assays also yield information both about the stability of the characterized RNA and the structural dynamics inherent to the characterized sequence. The name of the probes have been changed to P1 and RBS to reflect their target sequence (Supplementary Table 5).

REVIEWERS' COMMENTS:

Reviewer #3 (Remarks to the Author):

I have read the revised manuscript from Chauvier et al. and find that the authors have done a good job of revising the manuscript in response to the reviewers' suggestions. My scientific concerns with the description of the results have been resolved by changes to the text in the revision, and I think the manuscript is now suitable for publication.